# Exogenous Trehalose Improves Growth, Glycogen and Poly-3-Hydroxybutyrate (PHB) Contents in Photoautotrophically Grown *Arthrospira platensis* under Nitrogen Deprivation

**DOI:** 10.3390/biology13020127

**Published:** 2024-02-18

**Authors:** Nat-Anong Mudtham, Authen Promariya, Chanchanok Duangsri, Cherdsak Maneeruttanarungroj, Suchanit Ngamkala, Nattaphong Akrimajirachoote, Sorawit Powtongsook, Tiina A. Salminen, Wuttinun Raksajit

**Affiliations:** 1Program of Animal Health Technology, Department of Veterinary Nursing, Faculty of Veterinary Technology, Kasetsart University, Bangkok 10900, Thailand; nat.mudtham@gmail.com (N.-A.M.); authen.pr@ku.th (A.P.); cvtsnn@ku.ac.th (S.N.); 2Department of Animal Science, Faculty of Agricultural Technology and Agro-Industry, Rajamangala University of Technology Suvarnabhumi, Phranakhon Si Ayutthaya 13000, Thailand; chanshanok9259@gmail.com; 3Department of Biology, School of Science, King Mongkut’s Institute of Technology Ladkrabang, Bangkok 10520, Thailand; cherdsak.ma@kmitl.ac.th; 4Department of Physiology, Faculty of Veterinary Medicine, Kasetsart University, Bangkok 10900, Thailand; fvetnpa@ku.ac.th; 5Center of Excellence for Marine Biotechnology, Department of Marine Science, Faculty of Science, Chulalongkorn University, Bangkok 10330, Thailand; sorawit@biotec.or.th; 6National Center for Genetic Engineering and Biotechnology (BIOTEC), Pathum Thani 12120, Thailand; 7Structural Bioinformatics Laboratory, Biochemistry, Faculty of Science and Engineering, Åbo Akademi University, Tykistökatu 6 A, FI-20520 Turku, Finland; tiina.salminen@abo.fi

**Keywords:** *Arthrospira platensis*, *glgA*, *phaC*, PHA synthase, PHB, trehalose

## Abstract

**Simple Summary:**

Trehalose is an easily and economically obtainable non-reducing disaccharide, readily taken up by living organisms upon external application. *Arthrospira platensis* grew photoautotrophically under nitrogen-deficient conditions supplied with the appropriate trehalose, promoting glycogen and PHB production. This is supported by a notable increase in the upregulation of *glgA* and *phaC* expression, along with the induction of enzyme activity, including glycogen synthase encoded by *glgA* and PHA synthase encoded by *phaC*. The present study offers new insights into the contribution of exogenous trehalose in the regulation of glycogen and PHB production in *A. platensis*.

**Abstract:**

Glycogen and poly-3-hydroxybutyrate (PHB) are excellent biopolymer products from cyanobacteria. In this study, we demonstrate that nitrogen metabolism is positively influenced by the exogenous application of trehalose (Tre) in *Arthrospira platensis* under nitrogen-deprived (−N) conditions. Cells were cultivated photoautotrophically for 5 days under −N conditions, with or without the addition of exogenous Tre. The results revealed that biomass and chlorophyll-*a* content of *A. platensis* experienced enhancement with the addition of 0.003 M and 0.03 M Tre in the −N medium after one day, indicating relief from growth inhibition caused by nitrogen deprivation. The highest glycogen content (54.09 ± 1.6% (*w*/*w*) DW) was observed in cells grown for 2 days under the −N + 0.003 M Tre condition (*p* < 0.05), while the highest PHB content (15.2 ± 0.2% (*w*/*w*) DW) was observed in cells grown for 3 days under the −N + 0.03 M Tre condition (*p* < 0.05). The RT-PCR analysis showed a significant increase in *glgA* and *phaC* transcript levels, representing approximately 1.2- and 1.3-fold increases, respectively, in *A. platensis* grown under −N + 0.003 M Tre and −N + 0.03 M Tre conditions. This was accompanied by the induction of enzyme activities, including glycogen synthase and PHA synthase with maximal values of 89.15 and 0.68 µmol min^−1^ mg^−1^ protein, respectively. The chemical structure identification of glycogen and PHB from *A. platensis* was confirmed by FTIR and NMR analysis. This research represents the first study examining the performance of trehalose in promoting glycogen and PHB production in cyanobacteria under nitrogen-deprived conditions.

## 1. Introduction

Trehalose is a non-reducing disaccharide which is composed of two glucose units linked by an α, α-1,1-glycosidic linkage [1,2]. Trehalose can improve the growth and productivity of plants, as observed under varying water availability. Trehalose is easily and affordably accessible, and when externally applied, plants readily uptake it [3]. In addition, trehalose is a compatible solute able to respond to various stresses such as high temperature, drought, and osmotic stress [4]. Extensive research has been conducted on the role of trehalose in conferring resistance to desiccation stress in cyanobacteria. *Nostoc commune* Vaucher ex Bornet and Flahault, 1888, a terrestrial cyanobacterium, exhibits remarkable survival in a desiccated state. When faced with drought conditions, *N. commune* not only produces extracellular polysaccharides, crucial for desiccation tolerance, but also accumulates trehalose [5]. Similarly, other cyanobacteria such as *Phormidium autumnale* Gomont, 1892, and *Chroococcidiopsis* Geitler, 1933, have been observed to accumulate trehalose and sucrose in response to drought stress [6]. The synthesis of trehalose is triggered in response to water loss during the desiccation process. Conversely, when water becomes available and cells undergo rehydration, the trehalose content within the cells decreases [5]. Likewise, *Anabaena variabilis* Kützing ex Bornet and Flahault, 1886, and *Nostoc punctiforme* Hariot, 1891, accumulate trehalose under dehydration tolerance, but the quantities are not sufficiently high to provide protection as a molecular chaperone [7]. The mechanism through which trehalose alleviates stress conditions is closely linked to its capacity to stabilize membranes and protein structures [8]. The concentration of trehalose in the cells of *Spirulina* (*Arthrospira*) *platensis* Gomont, 1892, experienced a rapid increase when a high concentration of NaCl was introduced into the culture medium [9]. However, there are no reports on the results of trehalose on cyanobacterial performance under nutrient limitations. Even if the nutrient limitation promotes glycogen and polyhydroxybutyrate (PHB) levels in the cyanobacterial cells, the main drawback is the reduction in biomass production [10,11,12,13]. This suggests that the nitrogen metabolism is closely integrated with carbon metabolism across various aspects of cell function, including photosynthesis. Application of trehalose can potentially modify this regulation to prioritize the maintenance of productive functions under low nitrogen conditions [3]. Trehalose can be stored as a carbohydrate reserve and energy source.

*Arthrospira platensis* is a filamentous photosynthetic cyanobacterium that thrives in high alkalinity (pH 9 to 10) and high salinity [14,15]. Under stress situations, the strategic utilization of *A. platensis* has been developed as a resource for compatible solutes [9] and the production of common biopolymers [16], including glycogen [17] and PHB [10,18]. Altering the levels of trehalose in living organisms could serve as a method to enhance performance under nitrogen-limiting conditions [3,19]. Applying trehalose under low-nitrogen conditions resulted in an increase in the assimilation of both nitrate and ammonium, leading to a modified accumulation of amino acids [3]. This stimulation was correlated with enhanced photosynthesis, growth, and biomass of the plants and cyanobacteria.

The current study was carried out to evaluate the efficacy of exogenous trehalose in triggering *A. platensis* growth, as well as its impact on glycogen and PHB contents under nitrogen-deficient conditions. Furthermore, we investigated the expression of *glgA* (which encodes glycogen synthase) and *phaC* (which encodes PHA synthase) genes, crucial for the biosynthesis of glycogen and PHB, respectively. Additionally, we conducted quantitative and qualitative analyses of glycogen and PHB extracted from *A. platensis*.

## 2. Materials and Methods

### 2.1. Bacterial Strain and Culture Conditions

The culture of *Arthrospira platensis* IFRPD1182 was obtained from the Institute of Food Research and Product Development, Kasetsart University (IFRPD). The strain was pre-cultivated in a 250-mL Erlenmeyer flask with 50 mL of Zarrouk medium (pH 10.0) containing NaNO_3_ as a source of nitrogen. The cultures were incubated aerobically under continuous white-fluorescent illumination of 40 µmolE m^−2^ s^−1^ on a rotary shaker at 120 rpm and 32 °C with atmospheric CO_2_ for 7 days. After pre-cultivation, the cells were harvested and inoculated into 50 mL of nitrogen-deprived Zarrouk medium (−N) followed by cultivation under continuous illumination for various times up to 5 days. Different concentrations of trehalose were added to the −N medium when required. Biomass production and chlorophyll-*a* determination were examined according to [10,20], respectively.

### 2.2. Transcriptional Expression Study

Cells of *A. platensis* were harvested and immediately frozen in liquid nitrogen. Total RNA was extracted using TRIZol^TM^ reagent (Invitrogen, Waltham, MA, USA), following the manufacturer’s instructions. RNA was treated with RNase-free DNase (Promega, Madison, WI, USA) to remove contaminating DNA. The yield and purity of extracts were quantified using a NanoDrop spectrophotometer (Thermo Scientific, Waltham, MA, USA). Reverse transcription of 100 ng of total RNA per 20 µL reaction was carried out using the RevertAid First Strand cDNA Synthesis Kit following the manufacturer’s instructions (Thermo scientific, Waltham, MA, USA). The 286-bp of *glgA*, 340-bp of *phaC*, and 284-bp of *16S rRNA* of *A. platensis* were amplified using PCR with specific primer pairs, FglgA: 5′-TACGGACTCAGCCGAGAGTT-3′; and RglgA: 5′-GAAGGCAAACCGCATATTGT-3′ for *glgA*; FphaC: 5′-CCCGAAGCCGTAGATATTGA-3′; and RphaC: 5′-CTTTTTCGGCGTAGAGGTTG-3′ for *phaC*; and F16S: 5′-GTTTACGGGATTGGCTCAGA-3′; and R16S: 5′-TCTTGGTGAAAGCCGAGAGT-3′ for *16S rRNA*. The RT-PCR conditions were as follows: initial denaturation step at 94 °C for 5 min, followed by 25–33 cycles of denaturation at 94 °C for 45 s, then annealing at 50 °C for 45 s, and extension at 72 °C for 45 s. A final extension step was carried out at 72 °C for 10 min followed by reaction storage at 4 °C. The products were separated by electrophoresis on a 1.2% agarose gel and visualized using a gel imaging system (Omega Fluor^TM^, San Francisco, CA, USA). Relative quantification of target genes in each sample was normalized to the internal housekeeping gene, *16S rRNA,* under the same condition, which was represented as a relative transcript ratio (fold).

### 2.3. Glycogen Synthase Activity Assay

The activity of the glycogen synthase (GS) was determined from NADH oxidation at 30 °C, according to [21]. This assay was carried out in the reaction mixture (100 µL) containing 200 mM HEPES buffer (pH 7.0), 4 mg mL^−1^ glycogen, 2 mM ADPG (adenosine-5′-diphosphoglucose disodium salt), 0.7 mM phosphoenolpyruvic acid, 0.6 mM NADH β-nicotinamide adenine dinucleotide, 50 mM KCl, 13 mM MgCl_2_, pyruvate kinase (7.5 U), lactate dehydrogenase (15 U), and an appropriate amount of crude enzyme. The GS activity was measured spectrophotometrically at 340 nm. One unit of enzyme activity was defined as the amount of enzyme catalyzing the consumption of 1 µmol of NADH in 1 min at 30 °C. Protein content was determined by a Bio-Rad protein assay kit using bovine serum albumin (Sigma-Aldrich, Burlington, MA, USA) as standard.

### 2.4. PHA Synthase Activity Assay

The amount of CoA (reaction product) produced by PHA synthase was estimated using DL-hydroxybutyryl coenzyme A (HB-CoA) as an artificial substrate [22]. The experiment was performed in a reaction mixture (1 mL) comprising 50 mM of DL-3-hydroxybutyryl coenzyme A (3HB-CoA) and 20 mM 5, 5-Dithio-bis (2-nitrobenzoic acid) (DTNB) in 100 mM Tris-HCl buffer (pH 8.0), and enzyme incubation was carried out at 30 °C for 1 min. The liberation of CoA was measured at 412 nm by a spectrophotometer. One unit of the enzyme activity was defined as the amount of enzyme that catalyzes the conversion of 1 µmol substrate into a product in 1 min.

### 2.5. Glycogen Extraction and Quantitative Analysis

Glycogen was extracted from dry cells by a modified method from [23]. Briefly, dry cells of *A. platensis* (0.5 g DW) were digested by 30% (*w*/*v*) KOH, incubated at 100 °C for 90 min, and subsequently placed on ice. To precipitate glycogen, 900 µL of cold ethanol was added and kept on ice for 2 h. The precipitate was recovered by centrifugation at 4 °C, 10,000× *g* for 15 min. Pellets were washed twice with 90% and 70% cold ethanol, respectively, and then dried at 60 °C in a heat block for 10 min. Each dried sample was reconstituted in 200 µL of water and centrifuged at 10,000× *g* for 5 min at 4 °C, and the supernatant was subjected to high-performance liquid chromatography (HPLC) (Waters Corp., Milford, MA, USA) analysis. The glycogen content was determined as described by [17]. Oyster glycogen (Sigma-Aldrich, Burlington, MA, USA) was used as the standard for glycogen quantification.

### 2.6. PHB Extraction and Quantitative Analysis

The quantification of PHB was analyzed by HPLC (Waters Corp., Milford, MA, USA) with 5 μm of InertSustain C18 reverse-phase column (I.D. 4.6 × 150 nm) equipped with an SPD-20A UV/VIS detector at 210 nm [10]. Dry cells (0.5 g DW) were boiled with H_2_SO_4_ for 1 h to hydrolyze the PHB polymer into crotonic acid. HPLC analysis was performed using a 20 µL sample injection. The isocratic solvent system was run at 60% (*v*/*v*) of 0.1% acetic acid and 40% (*v*/*v*) of acetonitrile with a flow rate of 0.6 mL min^−1^. The HPLC chromatogram of the sample was calculated from the regression equation derived from the PHB standard (Sigma-Aldrich, Burlington, MA, USA) curve.

### 2.7. Fourier Transform Infrared (FTIR) Spectroscopy

FTIR analysis was performed in a Nicolet 6700 spectrometer (Thermo Scientific Inc., Waltham, MA, USA) by recording 100 scans with a spectral width ranging from 500 to 4000 cm^−1^, at a spectral resolution of 4 cm^−1^. The functional group patterns were determined using FTIR and compared with Oyster glycogen (Sigma-Aldrich, Burlington, USA) or PHB standard (Sigma-Aldrich, Burlington, MA, USA).

### 2.8. ^1^H- NMR and ^13^C-NMR Analysis

The PHB samples were dissolved in deuterochloroform, CDCl_3_ (Sigma-Aldrich, Burlington, MA, USA), at a concentration of 10 mg mL^−1^. The chemical structure was characterized using ^1^H and ^13^C resonance frequencies. The spectra were acquired using a Bruker AVANCE III HD/OXFORD 500 MHz NMR spectrometer. Tetramethylsilane (Si(CH_3_)_4_) was employed as the internal shift standard.

### 2.9. Statistical Analysis

All experiments were performed in triplicate, and the results were presented as mean values ± standard error of mean (SEM). One-way ANOVA was performed followed by Tukey’s post hoc tests using the GraphPad Prism software (GraphPad Software, Version 5, San Diego, CA, USA). A *p* value of < 0.05 was considered statistically significant.

## 3. Results

### 3.1. Effects of Exogenous Trehalose (Tre) on Biomass and Chlorophyll-a Contents under Nitrogen Deprivation

The *A. platensis* was grown in the normal nutrient condition (+N) or nitrogen-deprived condition (−N) with or without exogenously added Tre under continuous illumination. The biomass and chlorophyll-*a* contents of *A. platensis* grown with deficient nitrogen decreased significantly on the first day compared to those grown with sufficient nitrogen (Figure 1a,b). After five days, the biomass and chlorophyll-*a* levels had dropped by about 70–75%, and the culture had turned yellowish green. Interestingly, biomass and chlorophyll-*a* contents of *A. platensis* were enhanced at 0.003 M and 0.03 M of Tre supplementation in the −N medium after one day. However, even at a concentration as high as 0.3 M of Tre, there was no observed improvement against the growth inhibition caused by −N.

### 3.2. Effects of Exogenous Trehalose (Tre) on Glycogen and PHB Contents under Nitrogen Deprivation

The *A. platensis* was cultured under the normal nutrient condition (+N) or the nitrogen-deprived condition (−N) under continuous illumination with or without exogenously added Tre. The glycogen content was significantly increased (54.09 ± 1.6% (*w*/*w*) DW) in *A. platensis* cells grown under −N condition supplied with 0.003 M Tre for two days (*p* < 0.05). Subsequently, the glycogen content declined, but it still remained higher than in cells under −N conditions without Tre (Figure 2a). Increasing the Tre concentration with either 0.03 M or 0.3 M in the cultures reduced the glycogen level compared to that at 0.003 M Tre on the same days. Moreover, the PHB content of autotrophically grown *A. platensis* under −N condition increased to 2.6 ± 0.3% (*w*/*w* DW) on the first day. Particularly, the −N conditions yielded higher levels of PHB accumulation than the +N conditions. Under the −N condition with either 0.003 M or 0.03 M in the cultures, the increased PHB contents ranging from 4.5 ± 0.4 to 15.2 ± 0.2% (*w*/*w* DW) were observed from day 1 to day 5 (Figure 2b). The highest PHB content (15.2 ± 0.2% (*w*/*w* DW)) was greatly increased in cells grown under the −N condition supplied with 0.03 M Tre for three days (*p* < 0.05), and the PHB contents declined after that day.

### 3.3. Effects of Exogenous Trehalose (Tre) on the Expression of glgA and phaC Genes under Nitrogen Deprivation

The highest relative expression of the *glgA* gene was significantly observed in cells grown under the −N condition supplied with 0.003 M Tre, while the *phaC* gene exhibited an increase under −N with 0.03 M trehalose supplementation. Both *glgA* and *phaC* genes increased by approximately 1.2-fold and 1.3-fold, respectively (*p* < 0.05), compared to cells grown in the −N condition without Tre (Figure 3a,b). Additionally, it is noted that the expression of *glgA* was detected at low levels, particularly at very high concentrations of Tre at 0.03 and 0.3 M. On the other hand, the expression of *phaC* consistently remained high across all concentrations of Tre.

### 3.4. Effects of Exogenous Trehalose (Tre) on Glycogen Synthase and PHA Synthase under Nitrogen Deprivation

The specific enzyme assays were performed for the glycogen synthase (GS) and PHA synthase (PS) by using crude extracts of *A. platensis* grown under the normal nutrient condition (+N) or the nitrogen-deprived condition (−N) with or without exogenously added Tre. The highest specific GS and PS activities were increased 1.3-fold and 1.5-fold compared to the −N condition, approximately 89.15 and 0.68 µmol min^−1^ mg protein^−1^ in cells grown in −N + 0.003 M Tre and −N + 0.03 M Tre, respectively (Figure 4).

### 3.5. Fourier Transform Infrared Spectroscopy Analysis

The glycogen from *A. platensis* cells (ApGly) grown under the nitrogen-deprived condition (−N) supplied with 0.003 M Tre was analyzed by FTIR spectroscopy as shown in Figure 5a. The intense band at 3241.71, 3265.25, and 3248.04 cm^−1^ of ApGly, glycogen standard, and D-glucose, respectively were assigned to the O–H stretching vibration. The ApGly showed prominent peaks at 998.31 cm^−1^ for the antisymmetric stretching vibrations of C–O corresponding to the peaks recorded for the pure glycogen standard at 996.79 cm^−1^, while D-glucose also showed a peak at 992.07 cm^−1^. The region of 1200–800 cm^−1^ was assigned to the stretching vibrations of the C–O and C–C groups. Also, the PHB from *A. platensis* cells (ApPHB) grown under −N condition supplied with 0.03 M Tre was analyzed by FTIR spectroscopy (Figure 5b). The unique characteristics of ApPHB were presented by the strong carbonyl group (C=O) at 1718.36 cm^−1^ and asymmetric C–O–C stretching vibration at 1267.80 cm^−1^. Other adsorption bands obtained at 1452.54 and 1451.49 cm^−1^ designated the −CH3 groups. The absorption peaks at 2929.12, 2978.85, and 3432.72 cm^−1^ were characteristic peaks of alkane (−CH) and hydroxyl (−OH) groups.

### 3.6. NMR Analysis

The ^1^H-NMR spectrum of the extracted PHB, dissolved in deuterochloroform, revealed distinct proton signals in the range of (δ) 1.267–5.276 ppm, encompassing the 3HB subunit (Figure 6a). Within the 3-hydroxybutyrate subunit, signals in the range of 5.236–5.276 ppm were observed for the asymmetric carbon (−CH), corresponding to the chiral carbon atom. The multiplet resonance in the range of 2.581–2.627 and 2.450–2.492 ppm indicated diastereotopic methylene (−CH_2_) protons. Additionally, the doublet signals at 1.267–1.279 ppm pointed to methyl protons (−CH_3_). The ^13^C-NMR spectrum displayed vibrating carbon signals in the range of (δ) 19.760 to 169.134 ppm. A signal at 169.134 indicated carboxylic carbon (–C=O), while the signal at 67.605 ppm indicated asymmetric carbon as −CH. The signal at 40.786 ppm represented −CH_2_. Vibrating signals at 19.760 ppm were attributed to methyl carbon (−CH_3_) (Figure 6b).

## 4. Discussion

Trehalose is a non-reducing disaccharide extensively distributed in diverse organisms, including bacteria, yeasts, fungi, and plants [1]. The accumulation of trehalose is involved in response to abiotic stress, including osmotic stress [24]. However, the significance of trehalose uptake by living cells remains uncertain, given that all organisms possess enzymes capable of synthesizing and maintaining optimal levels for their metabolism. There are a few scattered reports on trehalose accumulation and utilization in cyanobacteria in response to nitrogen availability, particularly in *A. platensis* [8,25,26]. In conditions of nitrogen deficiency, the treatment of appropriate exogenous trehalose resulted in increased levels of chlorophyll-*a* and biomass contents in *A. platensis*, which contributed to the partial alleviation of nitrogen deficiency. Likewise, the application of external trehalose can enhance plant growth and overall performance in conditions of limited nitrogen availability, marked by a significant increase in the upregulation of nitrate and ammonium assimilation [3]. Under conditions of low nitrogen availability, the total carbohydrate content, which includes sucrose, trehalose, and glucose, in *Microcystis* Kützing, 1846, cells was enhanced compared to *Microcystis* cultured under conditions of high nitrogen availability [27]. According to our results, glycogen accumulation (54.09 ± 1.6% (*w*/*w* DW) induced by exogenous trehalose (0.003 M) could alleviate the damage to *A. platensis* caused by nitrogen-deprived (−N) conditions for 2 days. Indeed, glycogen accumulation is an essential component of the nitrogen stress response in *Synechococcus elongatus* Nägeli, 1849 [28]. Similarly, as reported by Hasunuma et al. [23], after being subjected to −N conditions for 3 days, *A. platensis* showed a significant accumulation of glycogen in its cells, reaching 63.2% (*w*/*w* DW). These findings suggest that trehalose plays a vital role in the glycogen accumulation in cyanobacteria under −N conditions, though the precise mechanism of trehalose synthesis in cyanobacteria remains not fully understood [24]. Additionally, the accumulation of glycogen may be beneficial during periods of starvation, providing both a stored source of energy and excess carbon [29]. Moreover, the flow of photosynthetically fixed carbon is redirected not only towards carbohydrates but also towards lipids in the metabolic pathway of protein synthesis [30].

The maximum PHB content, reaching 15.2 ± 0.2% (*w*/*w* DW), was observed in cells cultured under −N conditions and supplemented with 0.03 M trehalose for a duration of 3 days. However, PHB contents declined after this period. This pattern implies that the synthesis of PHB may be triggered by the concurrent presence of trehalose and a brief period of −N. One possible hypothesis is that PHB contents may arise from the induction of elevated intracellular NADPH levels, which is a prerequisite for the activity of the enzyme acetoacetyl-CoA reductase in the PHB biosynthetic pathway [31]. In addition to trehalose, cultures incubated with different carbon sources also showed an increase in PHB contents across different cyanobacterial species [32,33,34]. Remarkably, acetate yielded the highest PHB contents in *A. platensis* cells [10], surpassing those supplemented with butyrate, glucose, glycerol, and propionate under the studied conditions [10,32]. This emphasizes the influence of carbon sources on PHB accumulation in cyanobacterial cultures under −N conditions.

In the current study, it was found that the expression levels of *glgA* and *phaC* genes were increased in *A. platensis* grown under −N conditions and supplied with appropriated trehalose. Even when trehalose concentrations were as high as 0.03 and 0.3 M, the presence of *glgA* was hardly noticeable. In contrast, *phaC* expression was notably evident at the exceptionally high trehalose concentration of 0.3 M. These findings indicate a correlation between the increase in glycogen and PHB accumulation and the up-regulation of *glgA* and *phaC* transcripts in the presence of appropriated trehalose, particularly when *A. platensis* is cultivated under −N conditions. The altered response of *glgA* expression under −N conditions with trehalose supplementation might be attributed to an increase in carbon flux towards glycogen synthesis [17]. A comparable earlier study revealed up-regulation of the *glgA* and *glgC* genes in *Synechocystis* sp. PCC 6803 (*Synechocystis* Sauvageau, 1892) under nitrogen-deprived conditions [35].

In addition, the highest specific activities of glycogen synthase and PHA synthase increased by 1.3- and 1.5-fold, respectively, in cells grown under −N with 0.003 M and 0.03 trehalose supplementation, respectively, compared to cells in −N without trehalose. This, in turn, is anticipated to boost the production of PHB under −N conditions, especially when supplemented with trehalose. Nevertheless, at a concentration of 0.3 M trehalose, the activity of both enzymes decreased, despite the sustained high expression of the *phaC* gene, while *glgA* showed a notable reduction. This observation suggests that reduction of enzyme activity can also take place at the translation level, where the translation of a transcript can be influenced by a molecule that modifies the obstruction or accessibility of the ribosome binding site [36]. Moreover, the PHA synthase function has been previously elucidated in *Synechocystis* sp. PCC 6803, indicating mediation by the carbon-to-nitrogen (C/N) balance. This balance serves as a probable sink for reduction equivalents, thereby influencing PHB metabolism [37]. In *A. platensis*, the activity of PHA synthase relies on the existence of the catalytic triad, which comprises the amino acid residues cysteine, histidine, and aspartate, and it impacts PHB production [38].

The chemical structure and composition of glycogen and PHB from *A. platensis* was confirmed by FTIR and NMR analysis. Indeed, the chemical structure of carbohydrate bears a close resemblance to alcohol (O–H) functional groups [39]. The prominent bands observed at 3241.71 cm^−1^ for ApGly, 3265.25 cm^−1^ for the glycogen standard, and 3248.04 cm^−1^ for D-glucose were attributed to the O–H stretching vibration. Remarkably, the FTIR spectrum of D-glucose exhibited multiple bands covering the entire fingerprint region, and these bands were narrower and sharper compared to those observed for polysaccharides [40]. However, the most significant variations in the stretching vibration of C–O, as reported for the glycogen band, were observed in the 945-1060 cm^−1^ region [40,41]. In particular, ApGly displayed prominent peaks at 998.31 cm^−1^ for the antisymmetric stretching vibrations of C–O, corresponding closely to the peaks recorded for the pure glycogen standard at 996.79 cm^−1^, while D-glucose also exhibited a peak at 992.07 cm^−1^. The stretching vibration of C–O was associated with α-1, 4 glycosidic linkage regions in polysaccharides, while the monosaccharide was linked to the C–C group [40]. The results conclusively identified the extracted polymer as glycogen. In addition, the distinctive features of ApPHB were identified by the prominent carbonyl group (C=O) at 1718.36 cm^−1^ and the asymmetric C−O−C stretching vibration at 1267.80 cm^−1^, which are indicative of ester bondings in PHB polyesters. These patterns align with earlier reports [10]. Additionally, absorption bands observed at 1452.54 and 1451.49 cm^−1^ indicated the −CH_3_ groups, representing the stretching vibration for both ApPHB and the PHB standard. The absorption peaks at 2929.12, 2978.85, and 3432.72 cm^−1^ corresponded to characteristic peaks of alkane (−CH) and hydroxyl (−OH) groups, respectively, in ApPHB. The structure of extracted PHB obtained from this experiment closely resembles that of commercial PHB, characterized by the prevalence of ester, hydroxyl, methyl, and carbonyl groups.

NMR spectroscopy is employed for detailed structural elucidation [42,43,44]. The ^1^H NMR spectrum of PHB extracted from *A. platensis* revealed absorption bands for methyl (−CH_3_), methylene (−CH_2_), and asymmetric carbon (−CH) groups, confirming that the polymer structure was PHB. Additionally, ^1^H NMR has displayed multiplet signals (at 1.267–1.279 ppm, 2.450–2.627 ppm, and 5.236–5.276 ppm) indicating the presence of distinct hydrogen atom types (−CH_3_, −CH_2_, and −CH) relative to hydrogen bonds of PHB in the sample. The confirmation of the extracted polymer in the ^13^C NMR spectrum was based on the observed shifts in the signals of the carbonyl (–C=O), asymmetric carbon (−CH), methylene (−CH_2_), and methyl (−CH_3_) groups. The ^13^C NMR spectrum of the PHB produced by *A. platensis* showed chemical shift signals corresponding to the –C=O (169.134 ppm), −CH (67.605 ppm), −CH_2_ (40.786 ppm), and −CH_3_ (19.760 ppm) groups. The observed results are consistent with those reported for PHB produced in *Nostoc muscorum* Agardh ex Bornet and Flahault, 1886, [43] and *Anabaena* Bory ex Bornet and Flahault, 1886 [45].

## 5. Conclusions

The present study provides novel insights into the role of exogenous trehalose in mitigating the effects of nitrogen deprivation, as well as its regulatory impact on glycogen and PHB production in *A. platensis*. This is supported by a notable increase in the upregulation of *glgA* and *phaC* expression, along with the induction of enzyme activity, including glycogen synthase and PHA synthase, respectively, in *A. platensis* grown under nitrogen-deprived conditions and supplied with the appropriate trehalose. FTIR, ^1^H-NMR, and ^13^C-NMR analyses revealed that PHB was predominantly composed of butyric esters. Further investigation is needed to explore the metabolic interaction among trehalose, glycogen, and PHB accumulation in this aspect.

## Figures and Tables

**Figure 1 biology-13-00127-f001:**
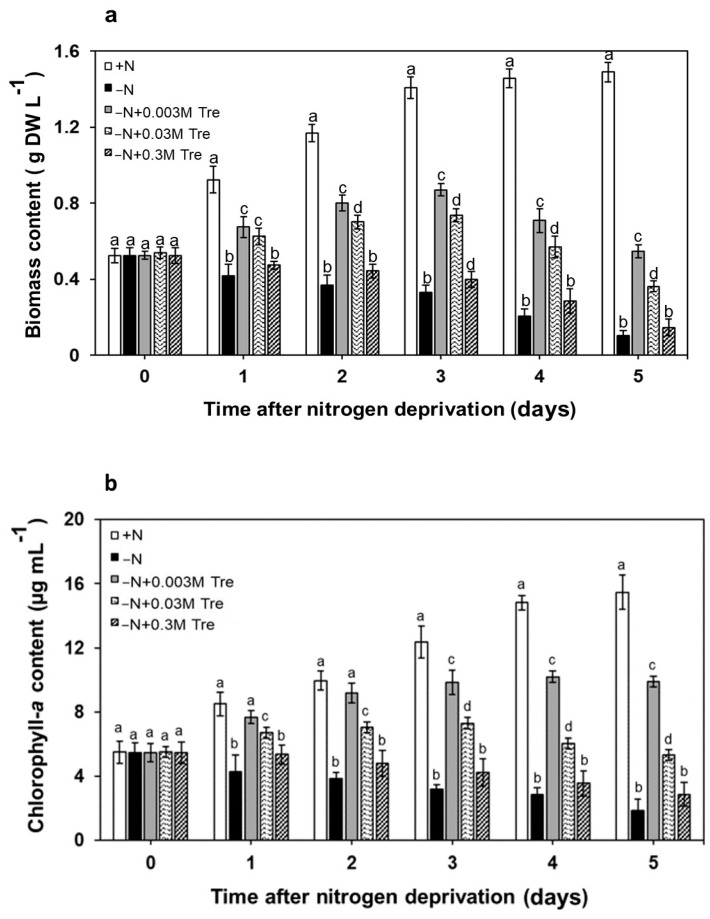
Biomass (**a**) and chlorophyll-*a* contents (**b**) of *A. platensis*. Cells were grown photoautotrophically for 5 days under normal nutrient condition (+N), nitrogen-deprived condition (−N), and −N supplied with trehalose (Tre) (−N + 0.003 M Tre, −N + 0.03 M Tre, or −N + 0.3 M Tre). Bar graphs represent mean values (±SEM) of three independent experiments. The letters a, b, c, and d indicate statistically significant difference between the group in each cultivation day (*p* < 0.05).

**Figure 2 biology-13-00127-f002:**
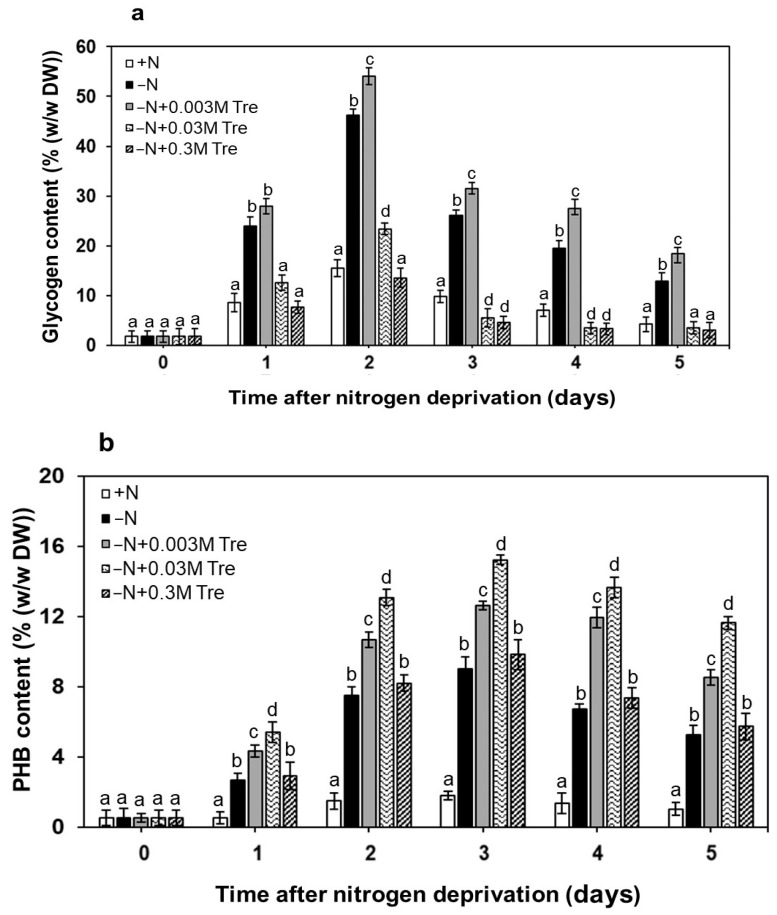
Glycogen (**a**) and PHB contents (**b**) of *A. platensis*. Cells were grown photoautotrophically under normal nutrient condition (+N), nitrogen-deprived condition (−N), and −N supplied with trehalose (Tre) (−N + 0.003 M Tre, −N + 0.03 M Tre, or −N + 0.3 M Tre). Bar graphs represent mean values (±SEM) of three independent experiments. The letters a, b, c, and d indicate statistically significant differences between the group in each cultivation day (*p* < 0.05).

**Figure 3 biology-13-00127-f003:**
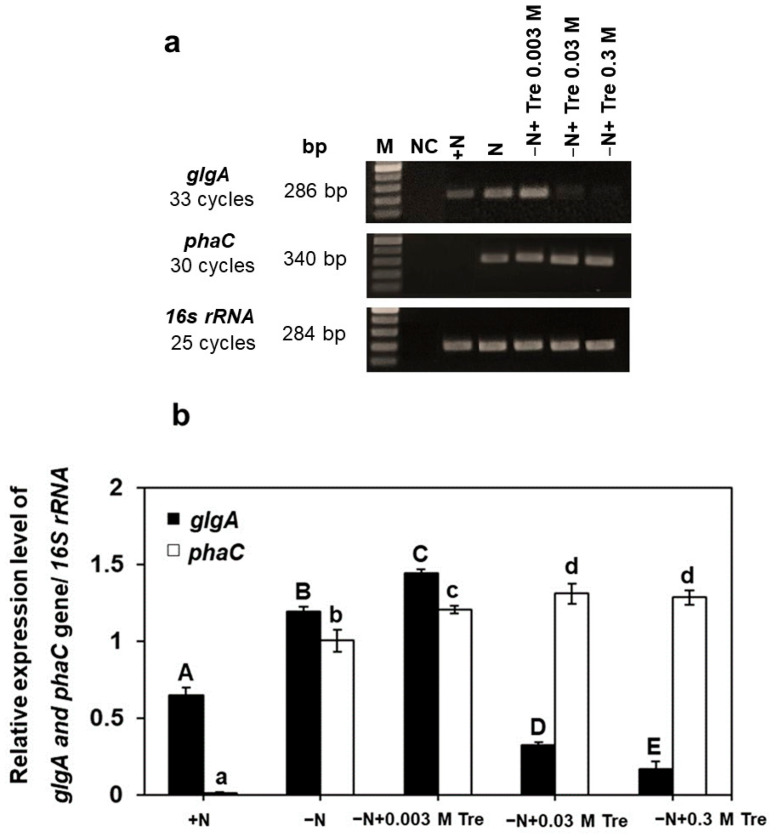
Transcriptional expression analysis of *A. platensis glgA* and *phaC* using RT-PCR. Cells were grown photoautotrophically under the normal nutrient condition (+N), nitrogen-deprived condition (−N), −N supplied with trehalose (Tre) (−N + 0.003 M Tre, −N + 0.03 M Tre, or −N + 0.3 M Tre), and no template control (NC). The upper panel (**a**) shows a typical example of DNA products resolved on an agarose gel, and the lower panel (**b**) shows the relative expression of *glgA* and *phaC* genes (mean values ± SD). The *16S rRNA* gene was used as a reference gene for normalizing the expression of target genes. The relative transcript levels of target genes were represented as the fold change. Bar graphs represent mean values of each data (± SEM) of three independent experiments. The letters A, B, C, D, E and a, b, c, d indicate statistically significant difference between the group in each cultivation day (*p* < 0.05).

**Figure 4 biology-13-00127-f004:**
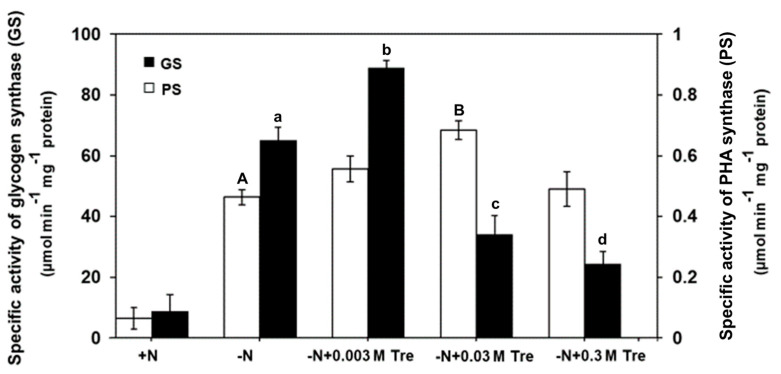
Specific activity determination of crude glycogen synthase (GS) and crude PHA synthase (PS) extracted from *A. platensis*. Cells were grown photoautotrophically under normal nutrient condition (+N), nitrogen-deprived condition (−N), and −N supplied with trehalose (Tre) (−N + 0.003 M Tre, −N + 0.03 M Tre, or −N + 0.3 M Tre). Bar graphs represent mean values (±SEM) of three independent experiments. The letters A, B and a, b, c, d indicate statistically significant difference between the group (*p* < 0.05).

**Figure 5 biology-13-00127-f005:**
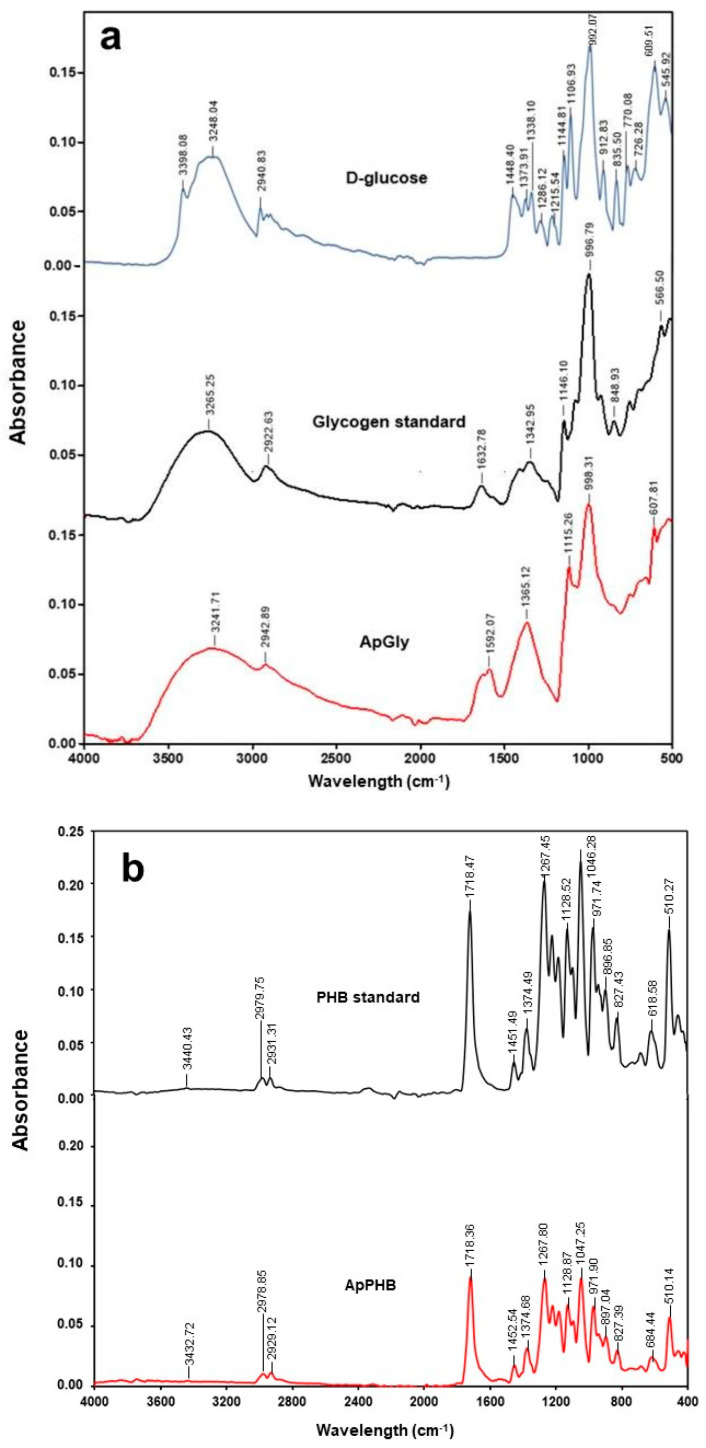
FTIR spectra determination of D-glucose (blue line) and *A. platensis* glycogen extracted (ApGly, red line) (**a**) and *A. platensis* PHB extracted (ApPHB, red line) (**b**). Cells were grown photoautotrophically under nitrogen deprivation (−N) supplemented with 0.003 M trehalose for glycogen or 0.03 M trehalose for PHB. FTIR spectra of samples were compared with commercial glycogen standard (black line) (**a**) or commercial PHB standard (black line) (**b**).

**Figure 6 biology-13-00127-f006:**
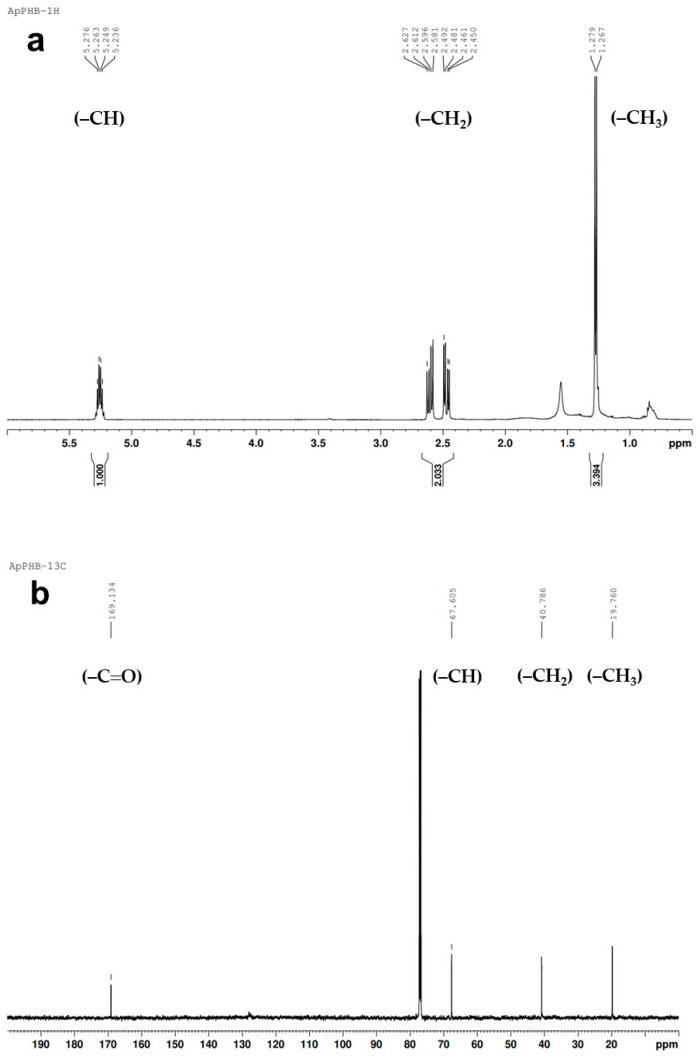
NMR spectrum of PHB extracted from *A. platensis.* (**a**) ^1^H NMR spectrum showing distinct proton signals in the range of (δ) 1.267–5.276 ppm and (**b**) ^13^C NMR spectrum showing distinct carbon signals in the range of (δ) 19.760 to 169.134 ppm.

## Data Availability

The original contributions presented in this study are included in the article; further inquiries can be directed to the corresponding author.

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
