# Peer review of "Exogenous Trehalose Improves Growth, Glycogen and Poly-3-Hydroxybutyrate (PHB) Contents in Photoautotrophically Grown Arthrospira platensis under Nitrogen Deprivation"

_biology, 2024, doi:10.3390/biology13020127_

Round 1

Reviewer 1 Report

Comments and Suggestions for Authors

The study is well designed  and done well. Subjected question, the effect of Trehalose on accumulation of glycogen and poly-3-hydroxybutyrate (PHB) is important and authors found appropriate answer on it. Manuscript can be accepted for publication.

Add: Authors addressed an important topic for cyanobacteria physiology: effect of extracellular additives to performance of alga culture during stress conditions. The Authors apply a valuable tool of scientific investigation methods, promote reliable experiments, found interesting results: improvement of survival and physiological state of Arthrospira platensis during Nitrogen starvation after supplementation of Exogenous trehalose and draw conclusions after that. Addressed topic. performance of cyanobacteria during nutrience deprivation stress is not well investigated yet, according to my knowledge. Authors provided sufficient introduction, material and methods, results and discussion sections in a manuscript. I think that the manuscript can be published as it is, without corrections.

Author Response

Dear Reviewer 1

Here is a point-by-point response to the reviewers’ comments and concerns.

Response to Reviewer 1

The study is well designed and done well. Subjected question, the effect of trehalose on accumulation of glycogen and poly-3-hydroxybutyrate (PHB) is important and authors found appropriate answer on it. Manuscript can be accepted for publication.

Add: Authors addressed an important topic for cyanobacteria physiology: effect of extracellular additives on performance of alga culture during stress conditions. The Authors apply a valuable tool of scientific investigation methods, promote reliable experiments, found interesting results: improvement of survival and physiological state of Arthrospira platensis during nitrogen starvation after supplementation of exogenous trehalose and draw conclusions after that. Addressed topic. performance of cyanobacteria during nutrient deprivation stress is not well investigated yet, according to my knowledge. Authors provided sufficient introduction, material and methods, results, and discussion sections in a manuscript. I think that the manuscript can be published as it is, without corrections.

Response: Your valuable comment is greatly appreciated. Thank you.

Reviewer 2 Report

Comments and Suggestions for Authors

This study offers new insights into the contribution of exogenous trehalose in the regulation of glycogen and PHB production in A. platensis.

The points below speak in favor of accepting the manuscript for the journal:

*The manuscript is clear, relevant for the field and well-structured.

*The abstract presented in the article characterizes the subject, reflects the purpose of the study, the main content and novelty of the article.

*The introduction contains historical and theoretical data according to modern literary sources.

I have only several minor comments. Please, see them below:

Line 28 – Please delete space between “(PH” and “B)”.

Line 49 – Please delete space between citation numbers e.g. «[1, 2]» and so on.

Lines 54, 58, 62, etc. – Please give the authority of each taxon.

Line 66 – It is better to write «Spirulina (Arthrospira) platensis».

Lines 86 – 93 – The paragraph needs to be rephrased or deleted as conclusion repetition in the introduction.

Line 313 - as reported by [23] > as reported by Hasunuma et al. [23].

Author Response

Dear Reviewer 2

Response to Reviewer 2

This study offers new insights into the contribution of exogenous trehalose in the regulation of glycogen and PHB production in A. platensis. The points below speak in favor of accepting the manuscript for the journal:

*The manuscript is clear, relevant for the field and well-structured.

*The abstract presented in the article characterizes the subject, reflects the purpose of the study, the main content and novelty of the article.

*The introduction contains historical and theoretical data according to modern literary sources.

I have only several minor comments. Please, see them below:

Thank you for pointing this out. We have made corrections according to your suggestion.

  1. Line 28 – Please delete space between “(PH” and “B)”.

Response: We have deleted space between “(PH” and “B)” in Line 28.

  1. Line 49 – Please delete space between citation numbers e.g. «[1, 2]» and so on.

Response: We have deleted space between citation numbers e.g. «[1, 2]» in Lines 49, 72, 79, 81, 83, 460, 490, 492, 530, 546.

  1. Lines 54, 58, 62, etc. – Please give the authority of each taxon.

Response: We have adjusted the scientific name as suggested in Lines 54, 55, 58, 59, 63, 64, 68, 467, 472, 505, 557, 558.

Nostoc commune Vaucher ex Bornet & Flahault, 1888,

Phormidium autumnale Gomont, 1892

Chroococcidiopsis Geitler, 1933

Anabaena variabilis Kützing ex Bornet and Flahault, 1886

Nostoc punctiforme Hariot, 1891

Spirulina (Arthrospira) platensis Gomont, 1892

Microcystis Kützing, 1846

Synechococcus elongatus Nägeli, 1849

Synechocystis Sauvageau, 1892

  1. Line 66 – It is better to write «Spirulina (Arthrospira) platensis».

Response: We have edited the name to Spirulina (Arthrospira) platensis as suggested in Line 68.

  1. Lines 87 – 92 – The paragraph needs to be rephrased or deleted as conclusion repetition in the introduction.

Response: We have rephrased the paragraph as suggested in Lines 88-93.

The current study was carried out to evaluate the efficacy of exogenous trehalose in triggering A. platensis growth, as well as its impact on glycogen and PHB contents under nitrogen-deficient conditions. Furthermore, we investigated the expression of glgA (which encodes glycogen synthase) and phaC (which encodes PHA synthase) genes, crucial for the biosynthesis of glycogen and PHB, respectively. Additionally, we conducted quantitative and qualitative analyses of glycogen and PHB extracted from A. platensis.

  1. Line 313 - as reported by [23] > as reported by Hasunuma et al. [23].

Response: We have made correction as suggested in Line 473.

Reviewer 3 Report

Comments and Suggestions for Authors

This study evaluates the use of trehalose to obtain PHB from Arthrospira platensis under nitrogen starvation conditions.

Although the topic is relevant, there are many studies that have evaluated the obtaining of PHA and PHB from cyanobacteria. In fact, I invite the authors to clearly establish the originality of this study.

Furthermore, the aim of the work is not very precise, here too I invite the authors to revise the aim subsection (lines 86-93).

The figures require improvement in quality, and I also recommend improving the legend of figures 1 and 2 because they are very small and difficult to read.

The part relating to the discussion of the NRM analysis was not discussed. Why was this analysis carried out? what does it say differently from FTIR analysis?

These questions should be clearly clarified in the paper.

Other remarks can be found in the file attached

Comments on the Quality of English Language

Minor revision of english

Author Response

Dear Reviewer 3

Response to Reviewer 3

This study evaluates the use of trehalose to obtain PHB from Arthrospira platensis under nitrogen starvation conditions. Although the topic is relevant, there are many studies that have evaluated the obtaining of PHA and PHB from cyanobacteria.

  1. In fact, I invite the authors to clearly establish the originality of this study.

Response: We have established the originality in this current study.

  1. Furthermore, the aim of the work is not very precise, here too I invite the authors to revise the aim subsection (lines 86-93).

Response: We have revised the paragraph outlining the aim as suggested in Lines 88-93.

The current study was carried out to evaluate the efficacy of exogenous trehalose in triggering A. platensis growth, as well as its impact on glycogen and PHB contents under nitrogen-deficient conditions. Furthermore, we investigated the expression of glgA (which encodes glycogen synthase) and phaC (which encodes PHA synthase) genes, crucial for the biosynthesis of glycogen and PHB, respectively. Additionally, we conducted quantitative and qualitative analyses of glycogen and PHB extracted from A. platensis.

  1. The figures require improvement in quality, and I also recommend improving the legend of Figures 1 and 2 because they are very small and difficult to read.

Response: We have adjusted the figure quality as suggested.

  1. The part relating to the discussion of the NRM analysis was not discussed. Why was this analysis carried out? What does it say differently from FTIR analysis?

Response: We have included further discussion for NMR analysis in Lines 546-558. FTIR is often used for analysis of functional groups, while NMR is employed for detailed structural elucidation.

Other remarks can be found in the file attached. Thank you for pointing this out.

  1. Line 89 Double check the sentence

Response: We have modified the sentence as suggested.

  1. Line 191 check the word "greatly"

Response: We have changed the word “greatly” to “significantly” in Line 191.

  1. Line 196 check the word “alleviation”

Response: We have modified the sentence as suggested in Lines 195-197, and we have changed the word “alleviation” to “improvement”.

  1. Line 197 Figure 1, increase the figure quality, and remove “a” and “b” below the figure as they are already mentioned inside the graph

Response: We have adjusted the Figure 1 quality and removed “a” and “b” below the figure.

  1. Line 303 check the word “an elevation of”

Response: We have changed the word “an elevation of” to “increased levels of” in Line 462.

  1. Line 307-309 rephrase the sentence for clarity

Response: We have rephrased the sentence as suggested in Lines 466-469.

Under conditions of low nitrogen availability, the total carbohydrate content, which includes sucrose, trehalose, and glucose, in Microcystis Kützing, 1846 cells was enhanced compared to Microcystis cultured under conditions of high nitrogen availability [27].

  1. Line 313-315 rephrase the sentence for clarity

Response: We have rephrased the sentence as suggested in Lines 473-475.

            Hasunuma et al. [23], after being subjected to −N conditions for 3 days, A. platensis showed a significant accumulation of glycogen in its cells, reaching 63.2% (w/w DW).

  1. Line 326 clarify “this stimulation”

Response: The word “this stimulation” refers to PHB contents. We have replaced the word “this stimulation” to “PHB contents” in Line 486.

  1. Line 329 In this study only trehalose was evaluated, if you are referring to scientific literature this should be rephrase

Response: We have rephrased the sentence as suggested in Lines 488-490.

In addition to trehalose, cultures incubated with different carbon sources also showed an increase in PHB contents across different cyanobacterial species [32,33,34]. Remarkably, acetate yielded the highest PHB contents in A. platensis cells [10], surpassing those supplemented with butyrate, glucose, glycerol, and propionate under the studied conditions [10,32].

  1. Line 359-360 rephrase the sentence for clarity

Response: We have rephrased and modified the sentence as suggested in Lines 519-521.

In A. platensis, the activity of PHA synthase relies on the existence of the catalytic triad, which comprises the amino acid residues cysteine, histidine, and aspartate, and it impacts PHB production [38].

  1. Line 385-387 There are many discussions regarding the FTIR analysis, but the discussion on NMR results is missing

Response: We have included further discussion for NMR analysis in Lines 546-558.

NMR spectroscopy is a primary method for efficiently detecting biopolymer structures [42, 43, 44]. The 1H NMR spectrum of PHB extracted from A. platensis revealed absorption bands for methyl (−CH3), methylene (−CH2), and asymmetric carbon (–CH) groups, confirming that the polymer structure was PHB. Additionally, 1H NMR has displayed multiplet signals (at 1.267–1.279 ppm, 2.450–2.627 ppm, and 5.236–5.276 ppm) indicating the presence of distinct hydrogen atom types (−CH3, −CH2, and –CH) relative to hydrogen bonds of PHB in the sample. The confirmation of the extracted polymer in the 13C NMR spectrum was based on the observed shifts in the signals of the carbonyl (–C=O), asymmetric carbon (–CH), methylene (–CH2), and methyl (–CH3) groups. The 13C NMR spectrum of the PHB produced by A. platensis showed chemical shift signals corresponding to the –C=O (169.134 ppm), –CH (67.605 ppm), –CH2 (40.786 ppm), and –CH3 (19.760 ppm) groups. The observed results are consistent with those reported for PHB produced in Nostoc muscorum Agardh ex Bornet and Flahault, 1886 [43] and Anabaena Bory ex Bornet and Flahault, 1886 [45].

Round 2

Reviewer 3 Report

Comments and Suggestions for Authors

The authors answered all of my questions.

Comments on the Quality of English Language

Minor revision of English